# Momentum-independent magnetic excitation continuum in the honeycomb iridate $H_3LiIr_2O_6$

A. de la Torre [1] ✉, B. Zager [1], F. Bahrami [2], M. H. Upton[3], J. Kim[3], G. Fabbris [3], G.-H. Lee [4], W. Yang [4], D. Haskel [3], F. Tafti [2] & K. W. Plumb [1] ✉

Understanding the interplay between the inherent disorder and the correlated fluctuating-spin ground state is a key element in the search for quantum spin liquids. $H_3LiIr_2O_6$ is considered to be a spin liquid that is proximate to the Kitaev-limit quantum spin liquid. Its ground state shows no magnetic order or spin freezing as expected for the spin liquid state. However, hydrogen zero-point motion and stacking faults are known to be present. The resulting bond disorder has been invoked to explain the existence of unexpected low-energy spin excitations, although data interpretation remains challenging. Here, we use resonant X-ray spectroscopies to map the collective excitations in $H_3LiIr_2O_6$ and characterize its magnetic state. In the low-temperature correlated state, we reveal a broad bandwidth of magnetic excitations. The central energy and the high-energy tail of the continuum are consistent with expectations for dominant ferromagnetic Kitaev interactions between dynamically fluctuating spins. Furthermore, the absence of a momentum dependence to these excitations are consistent with disorder-induced broken translational invariance. Our low-energy data and the energy and width of the crystal field excitations support an interpretation of $H_3LiIr_2O_6$ as a disordered topological spin liquid in close proximity to bond-disordered versions of the Kitaev quantum spin liquid.

Quantum spin liquids (QSLs) encompass a rich family of phases of matter with dynamically fluctuating and long-range entangled spins at zero temperature[1,2]. One particularly important QSL realization is the Kitaev model (KQSL). This exactly soluble model consists of highly-frustrated bond directional Ising-like interactions between spin-1/2 moments on the two-dimensional honeycomb lattice and can host topologically protected fractionalized excitations[3]. Material realizations of this model have been proposed in strong spin-orbit coupled $4d/5d$ transition metal (TM) compounds with edge-sharing $TML_6$ ($L$ = O, Cl) octahedra in a honeycomb lattice[4]. In this case, the Heisenberg, $J$, and symmetric off-diagonal, $\Gamma$, exchange terms between $J_{eff} = 1/2$ pseudospins are predicted to be suppressed in favor of finite anisotropic exchange, $K$[5]. However, the most heavily studied candidate Kitaev materials, $\alpha$-$Li_2IrO_3$, $Na_2IrO_3$[6–8], and $\alpha$-$RuCl_3$[9,10], all magnetically order due to the existence of additional exchange interactions of comparable magnitude to $|K|$, because of non-cubic distortions of the $TML_6$ octahedra, and/or $4d/5d$ orbital extent[11,12]. Nonetheless, measurements of the excitation spectrum in these compounds have revealed high-energy magnetic continuum consistent with dominant Kitaev nearest-neighbor bond-directional interactions[10,13–16].

[1]Department of Physics, Brown University, Providence, RI 02912, USA. [2]Department of Physics, Boston College, Chestnut Hill, MA 02467, USA. [3]Advanced Photon Source, Argonne National Laboratory, Argonne, IL 60439, USA. [4]Advanced Light Source, Lawrence Berkeley National Laboratory, Berkeley 94720, USA. ✉e-mail: adlt@brown.edu; kemp_plumb@brown.edu

$H_3LiIr_2O_6$ is the most salient example of an emerging generation of Kitaev compounds. It is synthesized from the parent compound $\alpha$-$Li_2IrO_3$ through the replacement of inter-honeycomb layer Li with H[17]. This leads to a reduction of inter-$LiIr_2O_6$ layer coordination from octahedral to linear and a consequent modification of the intra-layer Ir−O−Ir bond angles and Ir−Ir bond distance. As a result, super-exchange pathways are modified with respect to those of $\alpha$-$Li_2IrO_3$ and a different magnetic state is expected in $H_3LiIr_2O_6$[17,18]. Temperature-dependent measurements of the magnetic susceptibility in $H_3LiIr_2O_6$ show no evidence for long-range magnetic order down to 5 mK, despite a Curie−Weiss temperature $\theta_{CW} \approx -105$ K, as confirmed by the NMR Knight shift[19,20]. The NMR relaxation rate rules out a spin glass in favor of dynamically fluctuating spins[19]. Raman spectroscopy[21] finds a continuum of magnetic excitations similar to that in the proximate Kitaev magnet $\alpha$-$RuCl_3$[22,23]. However, the observation of a non-zero NMR relaxation rate and $T^{-1/2}$ $C/T$ divergence indicate $E = 0$ spin excitations that are not consistent with expectations for a pure Kitaev QSL state[19]. Thus, the nature of the unconventional magnetic ground state and the associated collective excitations in $H_3LiIr_2O_6$ remains unknown.

Achieving a microscopic description of the magnetic state in $H_3LiIr_2O_6$ is complicated by disorder. Quasi-static H zero point motion[20,24,25], present even in crystallographic pristine samples, can lead to local variations of the intra-layer exchange interactions[26,27]. Furthermore, the heavily faulted stacking structure due to the linear inter-layer O−H−O coordination might also result on random intra-layer magnetic exchanges[17,18]. It remains to be understood how the random distribution of magnetic exchanges modifies the magnetic Hamiltonian to render a state with no long-range magnetic order or frozen moments but with low-temperature low-energy spin excitations. One possibility is that bond-disorder brings $H_3LiIr_2O_6$ away from the QSL regime[26,27] into a state where the formation of a long-range magnetically order phase is inhibited but with short-range correlations remnant of ordered states in $\alpha$-$Li_2IrO_3$ and $Na_2IrO_3$[13,15]. A second possibility to account for the absence of frozen moments is that bond-disorder suppresses Kitaev exchange in favor of a random distribution of nearest neighbor $J$, promoting the formation of a random valence bond quantum paramagnet[28]. In these two scenarios, the low-energy excitations observed in $H_3LiIr_2O_6$ could be explained by the presence of fluctuating unpaired spins that give rise to the $C/T$ scaling[29,30]. Within a third alternative, the thermodynamic observations in $H_3LiIr_2O_6$ can be explained by bond-disordered extensions of the pure ($J = 0$, $\Gamma = 0$) and extended Kitaev QSL (BD-KQSL). In this case, bond-disorder acts to pin a random distribution of flux degrees of freedom in the underlying KQSL and leads to the divergent low-energy density of states and associated $T^{-1/2}$ low-temperature specific heat[19,31-36]. We remark that the disordered Kitaev state is proximal to the pristine KQSL, but it represents a disordered phase of topological matter. All of these models are distinct states that account for the absence of frozen moments and can explain the thermodynamic observations in $H_3LiIr_2O_6$. However, they can be distinguished by their high-energy collective excitations that encode the dynamical spin-spin correlations, $S(\mathbf{q}, \omega)$. For example, the proximity to magnetically ordered phases might lead to correlations on longer length scales than the nearest-neighbor-only of the BD-KQSL[13,15,37,38]. Similarly, singlet-triplet excitations at the scale of $J$ in a random valence bond state lead to a characteristic momentum dependence of $S(\mathbf{q}, \omega)$, with a maximum of intensity at the zone boundary and lack of intensity at the zone center[39,40], distinct from the BD-KQSL[32,41]. A momentum-resolved spectroscopic measurement of the spin excitation spectrum is essential to distinguish between these two opposing views, discern the role of disorder, and reveal the magnetic ground state of $H_3LiIr_2O_6$.

The small volume of available $H_3LiIr_2O_6$ single crystals (see the Methods subsections "The sample growth and characterization" and "Resonant inelastic X-ray scattering (RIXS)"), high neutron absorption

cross-section of Ir, and the incoherent scattering from H inhibits inelastic neutron scattering (INS) from accessing the magnetic spectrum in this compound[10,42-44]. Although RIXS remains limited by state-of-the-art energy resolution comparable to the theoretically predicted magnitude of K, high-resolution RIXS experiments at the Ir $L_3$ edge have emerged as an alternative to access $S(\mathbf{q}, \omega)$ in $5d$ Mott insulators[13,14,16,45].

Here, we use RIXS to reveal that the low-energy collective magnetic excitations of $H_3LiIr_2O_6$ are characterized by a broad momentum-independent continuum. The temperature evolution of these excitations indicates that they are distinct from the paramagnetic state, but there is no evidence for short-range correlations of a remnant-ordered state. While the low temperature collective magnetic excitations of $H_3LiIr_2O_6$ are suggestive of calculations of $S(\mathbf{q}, \omega)$ for KQSL, an exhaustive mapping of reciprocal space reveals a lack of a momentum dependence suggesting that $H_3LiIr_2O_6$ is best understood as a proximal to a bond-disordered KQSL.

## Results
### Low-energy collective excitations
In Fig. 1a we show the RIXS intensity of single crystals of $H_3LiIr_2O_6$ at the Ir $L_3$ ($E_i = 11.215$ keV) with state-of-the-art energy resolution (full width half maximum FWHM = 20 meV) at high symmetry points of the pseudo-honeycomb lattice at $T = 10$ K following the experimental geometry sketched in Fig. 1b. Our measurement temperature is one order of magnitude smaller than $\theta_{CW}$ and below the ordering temperature of pristine $\alpha$-$Li_2IrO_3$[8]. The RIXS scans reveal a broad, $\sigma = 40 \pm 5$ meV bandwidth, inelastic signal centered around $E = 25 \pm 5$ meV. This signal resonates at the Ir $L_3$ edge, confirming that these excitations arise from a direct RIXS process and pointing to a magnetic origin[16] (see Supplementary Information) and is well described by an over-damped harmonic oscillator (DHO) weighted by a Bose factor ($1/(1 - e^{-E/k_B T})$) and convolved with a Gaussian resolution[46]. The tail of the DHO extends into a high-energy magnetic continuum spanning up to $E = 170$ meV. Given the absence of any sharp features over a bandwidth that is 8 times larger than the energy resolution of our measurement, we attribute this signal to a continuum of magnetic excitations. The inelastic spectrum is remarkably similar at all high symmetry points, being of equal or comparable intensity to the elastic line and without an evident dispersion or intensity modulation. The absence of a momentum dependence is contrary to expectations for a random distribution of singlets in a valence bond solid with dominant nearest neighbor correlations[39,40] and disordered Heisenberg AFM models[47] that predict a vanishing intensity at $\Gamma$ points. In Fig. 1d we show the RIXS spectra at the ordering vectors for the three 120° domains of the incommensurate spiral order in $\alpha$-$Li_2IrO_3$ $\mathbf{q}_\alpha = [0.16, -0.5]$, $[0.16, 0.5]$, $[0.32, 0]$[13]. The inelastic data across all momentum points and azimuthal angles (see Supplementary Information) can be fit to the same functional form as for the high symmetry points in Fig. 1a without any additional broadening or energy shift to the DHO, and without the need to account for the gapless acoustic magnon at $q_\alpha$ observed in $\alpha$-$Li_2IrO_3$ below $T_N = 15$ K[13].

### Temperature-dependent RIXS spectra
The temperature-dependent RIXS intensity at $\Gamma$ [Fig. 2a] is suggestive of a state that is intrinsically different from the high-temperature paramagnet. At room temperature, we find overdamped excitations with no discernible momentum dependence, as expected for the paramagnetic state [Fig. 2c] and consistent with previous high-temperature RIXS measurements in $Na_2IrO_3$ and $\alpha$-$Li_2IrO_3$[16]. Although the RIXS spectra of $H_3LiIr_2O_6$ are characterized by overdamped excitations at all temperatures, the detailed temperature dependence reveals that the excitations for $T \lesssim \theta_{CW} \approx 105$ K are distinct from those in the paramagnetic state. The DHO amplitude ($A$), monotonically increases as the temperature is lowered [square markers; Fig. 2b]. We

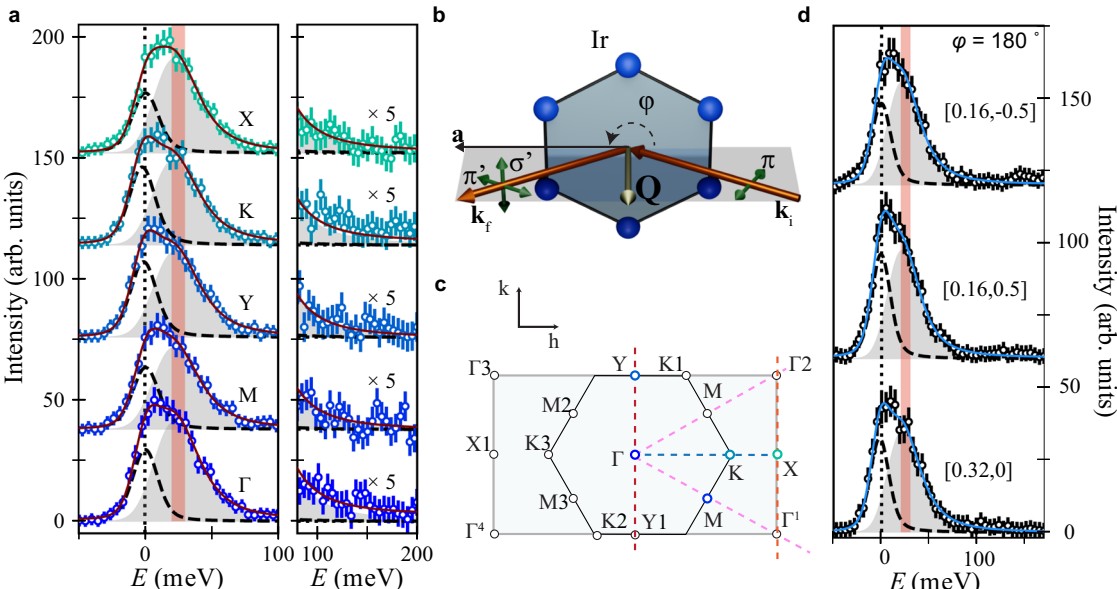

**Fig. 1 | High-resolution resonant inelastic X-ray spectra of $H_3LiIr_2O_6$ at the Ir $L_3$ edge. a** (left) RIXS spectra at $T = 10$ K at high symmetry points of the Brillouin zone (BZ) (circular markers). The magnetic continuum extends up to 150 meV as shown in the right panel (right; data are scaled by a factor of 5). Spectra have been shifted vertically for clarity. **b** Sketch of the scattering geometry. The hexagonal arrangement of blue spheres represents a honeycomb layer of $Ir^{4+}$ ions within the monoclinic crystal structure of $H_3LiIr_2O_6$. The incident ($\mathbf{k_i}$) and outgoing ($\mathbf{k_f}$) radiation (orange arrows) define the scattering plane (gray), with momentum transfer $\mathbf{Q}$ (silver arrow). Green arrows show the polarization of the X-ray electric field ($\pi$: in-plane; $\sigma,\sigma'$: out-of-plane for the incoming and outgoing X-ray beam, respectively). $\varphi$ is the azimuthal angle defined by the crystallographic a-axis and the scattering plane. **c** Schematic of the extended hexagonal BZ highlighting relevant symmetry points and directions (dashed lines follow the color scheme of Fig. 3) explored in this study. $L$ varies between 5.91 and 5.95 r.l.u. **d** RIXS intensity at the wavevectors of the 120° spiral order of $\alpha$-$Li_2IrO_3$. Solid lines in (**a**) and (**d**) are fit to the data, including a Voigt profile for the elastic line (dotted black line) and a damped harmonic oscillator (gray shading) centered at $E_0 = 25$ meV, red bar of width 10 meV reflects the statistical uncertainty in determining $E = 0$ meV. All data were taken at $\varphi = 180°$.

observe a softening of the center energy of the magnetic continuum from $E = 38 \pm 5$ meV with $\sigma = 45 \pm 5$ meV bandwidth at $T = 300$ K $\approx 3\theta_{CW}$ to $E = 25 \pm 5$ at $T < \theta_{CW}$ [circular markers; Fig. 2b]. Between $\theta_{CW}$ and $T \approx 200$ K the DHO is centered at $E = 30 \pm 5$ meV; this temperature range corresponds to a hump of 130 K centered at 200 K seen in magnetic susceptibility measurements[19] signifying the onset of nearest neighbor correlations in $H_3LiIr_2O_6$[48,49]. Note that the relatively coarse energy resolution of our measurements means that we cannot distinguish if this intensity and energy shift arises from the transfer of intensity between different modes above and below $\theta_{CW}$ or to a mode softening. While a direct comparison between the RIXS and Raman spin-flip process is complex, the evolution of the RIXS intensity with temperature resembles that seen in Raman spectroscopy measurements of Kitaev magnets[21,23,50,51].

## Momentum-independent excitations

One possible explanation for the excitations we observe is that bond-disorder and stacking faults act to inhibit an underlying magnetically ordered phase in $H_3LiIr_2O_6$[26,27]. In this scenario, remnant dynamical correlations reflecting a disorder limited short-range order should be present and give rise to a characteristic momentum dependence of $S(\mathbf{q}, \omega)$[52]. Good points of comparison are momentum-resolved measurements of magnetic X-ray scattering in $\alpha$-$Li_2IrO_3$ and $Na_2IrO_3$ above their magnetic transition temperatures. For both compounds, the diffuse magnetic scattering is characterized by an intensity modulation that peaks at their respective ordering vectors and depends on the projection of the incident X-ray polarization on spins thermally-fluctuating about an average orientation[13,15]. Similar information can be extracted by examining the magnetic RIXS response as a function of momentum and azimuthal angle $\varphi$ as sketched in Fig. 1c, d. We extract the inelastic RIXS intensity by subtracting the elastic line contribution determined from a fit to a resolution limited Voigt profile. Figure 2a

shows the momentum dependence of inelastic magnetic intensity along high symmetry directions covering multiple zone center ($\Gamma$) and zone boundary ($M$, $K$, $Y$, and $X$) points with $H \le 0$ and $\varphi = 0$. Equivalent points with $H \ge 0$ and $\varphi = 180°$ are shown in Fig. 2b. For both azimuthal configurations, the excitation spectra has comparable intensity at the zone center and zone boundary, it is non-dispersive, and devoid of sharp coherent modes. These data, together with Fig. 1, rule out any short-range spiral order correlations as seen in $\alpha$-$Li_2IrO_3$[13] or short-range zig-zag correlations as present in $Na_2IrO_3$ that lead to strong modulation of the magnetic diffuse scattering at $M$, $Y$, and $X$[14,15]. We thus rule out a disorder limited correlations remnant of an ordered state[52] and conclude that the spin excitations in $H_3LiIr_2O_6$ are intrinsically different to that of the parent compound or $Na_2IrO_3$. In Fig. 3a–d we plot the dependence of the center energy of the DHO along four additional high symmetry directions covering the totality of the BZ for three different values of $\varphi$. The inelastic intensity integrated after subtraction of the elastic line in the energy range $E \in [-30, 170]$ meV is shown in Fig. 3e–h. We observe no dispersion of the broad continuum centered at $E = 25 \pm 10$ meV (experimental resolution FHWM = 28 meV). The slight intensity modulation in our momentum-dependent data does not follow any of the structural symmetries of $H_3LiIr_2O_6$ and it is correlated variations in the elastic line intensity that are difficult to decouple given our experimental resolution (see Supplementary Information). As such, we assign it to an experimental artifact possibly related to self-absorption effects[16]. Our RIXS measurements covering many Brillouin zones, azimuthal angles, and the full energy bandwidth of magnetic excitations in $H_3LiIr_2O_6$ reveals dynamical correlations that are isotropic and with a featureless momentum dependence of the integrated intensity.

Our data demonstrates the existence of magnetic excitations at high energies (25 meV) in addition to the low-energy divergence observed in $C/T$ measurements in $H_3LiIr_2O_6$. The appearance of two

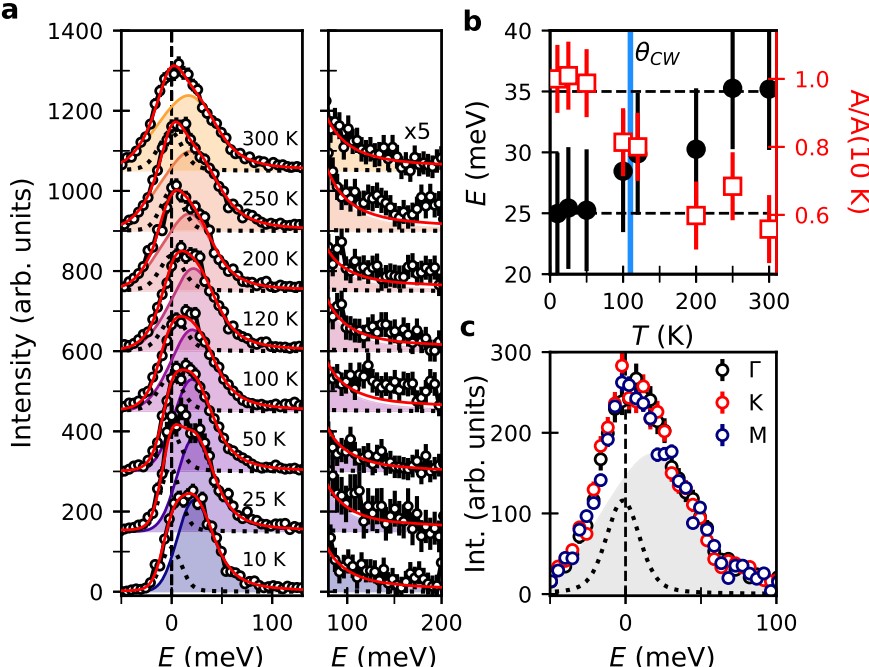

**Fig. 2 | Temperature evolution of the magnetic continuum in H₃LiIr₂O₆. a** High-resolution RIXS spectra at Γ (**Q** = [0, 0, 5.7]) and azimuthal angle $\varphi = 0°$, as a function of temperature. Spectra have been shifted vertically for clarity. **b** Circular markers indicate the center position of the damped harmonic oscillator (DHO) component as extracted from a fit to the data. Square markers are the extracted amplitude of the DHO normalized to the value at $T = 10$ K. The blue solid line indicates the Curie–Weiss temperature, $\theta_{CW}$, and the dashed black lines are guides to the eye. Black error bars are the systematic error in determining $E = 0$ meV. Red error bars are the statistical uncertainty of RIXS intensity. **c** $T = 300$ K RIXS intensity (circular markers) at three high symmetry points (Γ, black, $K$, red, $M$, blue). The solid line is a fit to the data including a Voigt profile for the elastic line (dotted black line) and a damped harmonic oscillator (shading).

energy scales is suggestive of calculations of $S(\mathbf{q}, \omega)$ for pure, extended, and bond-disorder gapless KQSL. Within these models the energy dependence of $S(\mathbf{q}, \omega)$ is characterized by an intense low-energy ($E \approx 0$) peak at the energy of the fractionalized fermionic excitations that decays exponentially into a high-energy continuum (up to $6J$)[32,37,38,41]. The momentum dependence shows broad isotropic excitations with an intensity modulation that depends on the sign of $K$. For an antiferromagnetic $K$ the $q$ dependence of the broad magnetic continuum at $E = |K|$ exhibits a shift of spectral weight from Γ to the zone boundary, while for the ferromagnetic (FM) Kitaev case a maximum of intensity is expected at Γ[37,38,41,52]. Calculations of the RIXS response for the Honeycomb Kitaev model distinguish between spin conserving (SC) and spin non-conserving (SNC) excitations channels which correspond to dispersive fractionalized fermionic excitations and bound gauge flux excitations, respectively. Within our experimental configuration ($\pi$ incident polarization; $2\theta = 90°$ scattering angle) the intensity is dominated by polarization-switching SNC channels leading to a nearly featureless excitation spectrum that resembles that accessible by inelastic neutron scattering measurements[53]. However, the lack of a clear peak of intensity at Γ and of any momentum dependence across the BZ is suggestive of broken translation invariance due to bond-disorder[52]. Thus, we assign the origin of the broad collective excitations centered at $E = 25 \pm 5$ meV and high-energy continuum extending up to $E = 170$ meV to the existence of dominant bond-dependent nearest-neighbor spin-spin correlations with a FM Kitaev exchange, $|K| \approx 25$ meV in a bond-disorderd background. While a random distribution of exchange values due to disorder hinder the exact determination of $K$, $|K| \approx 25$ meV, is comparable to that of $\alpha$-Li₂IrO₃ and Na₂IrO₃[13–16,54], to that extracted from the maximum of the continuum intensity in Raman scattering measurements at around $\omega \approx 40$ mev = $1.5K$ ($K \approx 26$ meV)[21], and to that from first principles calculations[26,27]. We remark that the energy resolution of our RIXS experiment $E = 20$ meV integrates over low-energy modes and prevents us from resolving

the $E \approx 0$ divergence predicted in the density of states for BD-KQSL models[32]. Similarly, we cannot exclude additional finite but comparable to $k_B T = 0.87$ meV $J$, Γ or any other exchange terms in the magnetic Hamiltonian of H₃LiIr₂O₆[26]. Moreover, the observed redistribution of RIXS intensity across the onset temperature for magnetic correlations in a Kitaev model provides further evidence for a low-temperature collective excitation spectrum that reflects Kitaev physics in H₃LiIr₂O₆ [Fig. 2]. Our RIXS data points to H₃LiIr₂O₆ hosting a unique disordered topological state in close proximity to the KQSL resulting from the interplay of $K$ and bond-disorder[32,52].

## Local Ir electronic structure

We finally comment on the specific effects of H on the local Ir electronic structure. While determining the position of the H atoms is not directly accessible in an X-ray experiment, measurements of the crystal field energies and width provide information about any local disorder on IrO₆ octahedra[55]. In Fig. 4a we show the RIXS response of H₃LiIr₂O₆ near the Ir $L_3$ as a function of incident energy, $E_i$, and energy transfer, $E$. We focus our discussion on the $0.25 < E < 1.2$ eV range corresponding to the intra-$t_{2g}$ excitations[56]. In this range, the RIXS spectrum is characterized by a set of four Gaussian peaks [Fig. 4b] (see Supplementary Information) of $FWHM_{A-D} = 75$–$120$ meV, larger but comparable to that observed for $\alpha$-Li₂IrO₃[56]. We model the crystal field excitations in H₃LiIr₂O₆ [Fig. 4b], by calculating the RIXS intensity from the exact diagonalization of a model Hamiltonian for an Ir⁴⁺ including nearest-neighbor hopping and local disorder[55,57]. Disorder is encoded by sampling the hopping parameters and the magnitude of the trigonal crystal field, $\delta$, from a normal distribution. The data is well described by including an O mediating hopping with mean value $t_O = 440$ meV. This value is well within the range of values extracted from density functional theory for an ideal $C2/m$ structure[26], and is 10% larger than that needed to account for the RIXS spectra of $\alpha$-Li₂IrO₃ (see the Methods subsection "Exact diagonalization calculations"). The larger

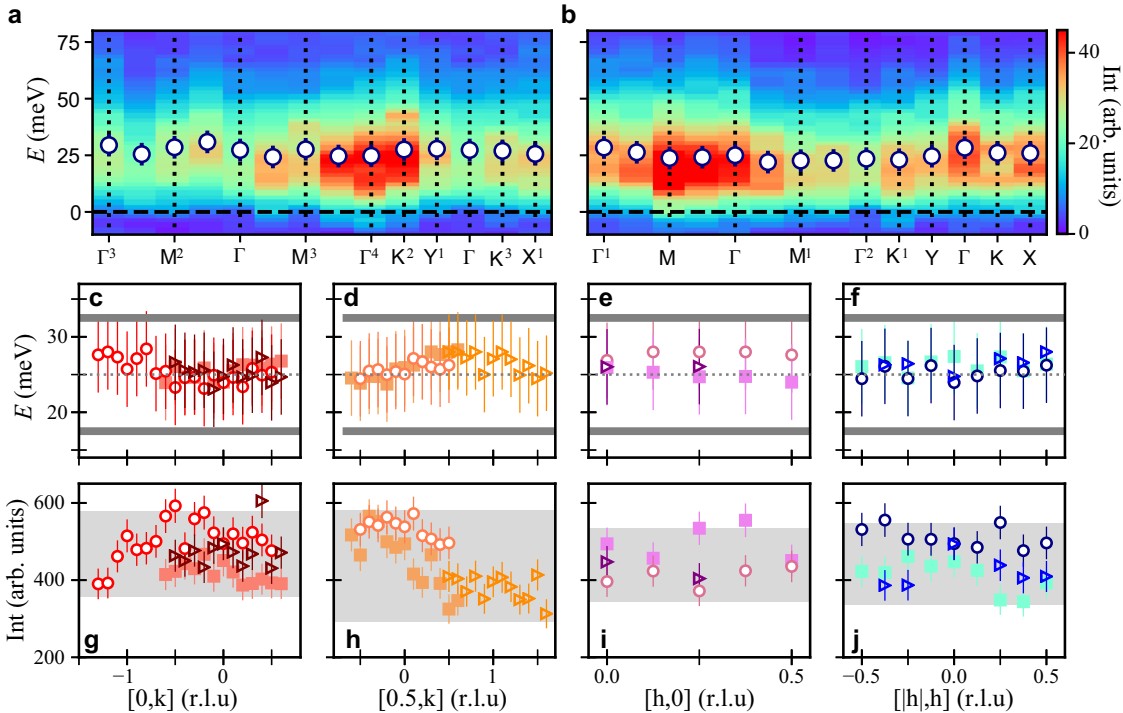

**Fig. 3 | Absence of short-range correlations in the momentum-independent magnetic continuum. a** High-resolution elastic-background-subtracted RIXS intensity along high symmetry paths with $H \leq 0$, $\varphi = 0°$ and, **b**, $H \geq 0$, $\varphi = 180°$. Overlaid circular markers indicate the center position of the continuum of magnetic excitations as extracted from a fit to the data. Error bars are the systematic error in determining $E = 0$ meV. The colorbar applies to both (**a**) and (**b**). **c**–**f** Center energy of the continuum of magnetic excitations across the Brillouin zone as extracted from a fit to the inelastic RIXS intensity to a damped harmonic oscillator.

The dashed line at $E = 25$ meV is the average momentum position of the continuum. The energy resolution was relaxed to a FWHM = 28 meV as indicated by the solid gray lines. Error bars are the systematic error in determining $E = 0$ meV. **g**–**j** Integrated elastic-background-subtracted RIXS intensity in the range $E \in [-50, 170]$ meV. The gray-shaded rectangle represents the variation of the elastic intensity. error bars are the statistical uncertainty of RIXS intensity. Circular markers ($\varphi = 0°$), triangular markers ($\varphi = 180°$), and square markers ($\varphi = 90°$) show the azimuthal dependence.

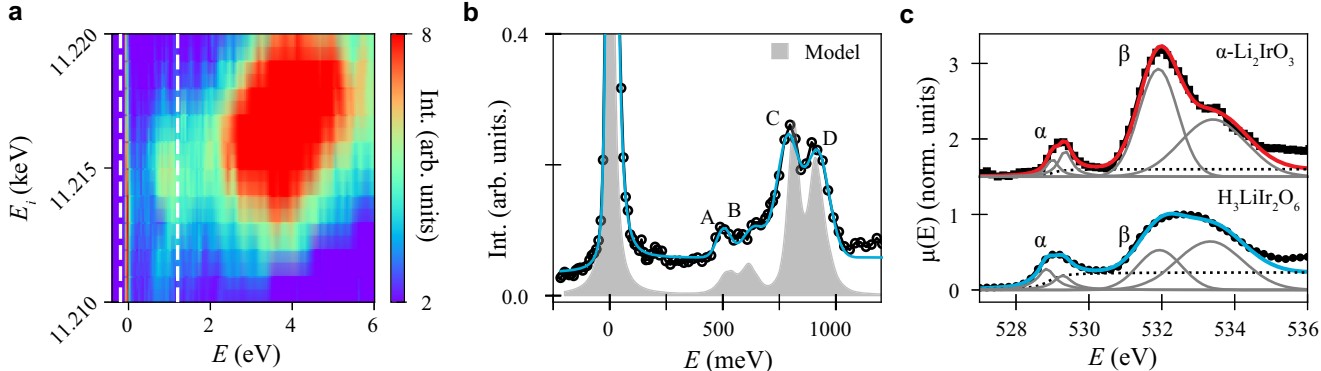

**Fig. 4 | Local Ir electronic structure. a** RIXS intensity as a function of the incident X-ray energy near the Ir $L_3$ edge. The intense inelastic feature centered at $E \approx 3.8$ eV is from transitions between occupied $t_{2g}$ states and empty $e_g$ orbitals in a Ir$^{4+}$ ion (see Supplementary Information). White dashed lines in (**a**) delineate the energy range shown in (**b**). **b** Intra-$t_{2g}$ RIXS excitations at $E_i = 11.215$ keV (circular markers) compared to the calculated RIXS intensity from exact diagonalization calculations. Blue solid line is a fit to the data including a Voigt peak, a DHO, four Gaussian peaks

and an arctan step to account for the background (A–D). **c** O K-edge X-ray absorption spectroscopy data ($\mu(E)$) and fit to the data in H$_3$LiIr$_2$O$_6$ (circular markers; blue line) and $\alpha$-Li$_2$IrO$_3$ (square markers; orange line). $\alpha$ and $\beta$ highlight the two main features in the XAS data as discussed in the main text. Solid gray lines show the two Lorentzian and two Gaussian profiles included in the fit. Dashed line represents the arc tangent step to account for the electron hole continuum.

value of $t_O$ is suggestive of an increased Ir $t_{2g}$-O $2p$ hybridization as a result of the introduction of $H$. This is substantiated by a qualitative comparison of the O-K edge XAS data between H$_3$LiIr$_2$O$_6$ and $\alpha$-Li$_2$IrO$_3$ (see the Methods subsection "O-K edge XAS"). The intensity and width of the $\alpha$-peak at $E = 529$ eV reflects the degree of hybridization between O $2p$ and $t_{2g}$ states[58], which is 1.16 times more intense and 1.4 times broader in H$_3$LiIr$_2$O$_6$ than in $\alpha$-Li$_2$IrO$_3$. Moreover, we find a comparable

value of the mean $\delta$ between both compounds ($\delta = -48$ meV in H$_3$LiIr$_2$O$_6$ and $\delta = -50$ meV in $\alpha$-Li$_2$IrO$_3$), in agreement with the average <2% change in the Ir–O bond angles and <4% in Ir–Ir bond distance with respect to that of $\alpha$-Li$_2$IrO$_3$[18]. However, we find that a large distribution of $\delta$ values, as encoded by the standard deviation $\sigma_\delta = 20$ meV, is needed to account for the broader crystal field excitations in H$_3$LiIr$_2$O$_6$ and the relative intensity ratio between peaks $A$–$B$ and $C$–$D$. This is

consistent with the existence of slow H-ion motion at low temperature, which can generate variations of the local $IrO_6$ environment[20,24]. Thus, our Ir crystal field RIXS data, O XAS measurements and analysis is consistent with the existence of local bond-disorder on $IrO_6$ octahedra and points to an enhanced Ir–O hybridization as the leading mechanism favoring a dominant Kitaev-like exchange in $H_3LiIr_2O_6$.

## Discussion

In summary, the spin excitation spectrum of $H_3LiIr_2O_6$ is characterized by broad isotropic excitations of comparable intensity to the elastic line without a momentum dependence of the integrated intensity. Altogether, our data support an interpretation of the magnetic excitation spectrum of $H_3LiIr_2O_6$ as emerging from the interplay of disorder and dominant Kitaev-like nearest-neighbor bond-directional interactions between dynamically fluctuating spins. Of the theoretical proposals that have been put forward to explain the thermodynamic properties of $H_3LiIr_2O_6$, bond-disorder KQSL models are the closest to describing our observations[32,52]. However, the lack of a clear intensity modulation across the BZ indicate that these excitations are of different nature to that of a pure KQSL despite being a magnetic state dominated by quantum fluctuations.

## Methods

### Sample growth and characterization

Precursor single crystals of $\alpha$-$Li_2IrO_3$ were grown as described elsewhere[59]. $40 \times 40\,\mu m$ single crystals of $H_3LiIr_2O_6$ were grown via a topotactic exchange by placing $\alpha$-$Li_2IrO_3$ in an acidic environment for several hours[18]. Sample characterization including powder X-ray diffraction, specific heat, and magnetization measurements can be found in ref. [17].

### Resonant inelastic X-ray scattering

RIXS measurements were performed at the 27 ID beamline of the Advance Photon Source using a horizontal scattering geometry with $\pi$-incident X-ray polarization. The low-energy RIXS spectra summing over $\pi$-$\sigma'$ and $\pi$-$\pi'$ polarization channels were collected with a fixed 2-m radius spherically diced Si(844) analyzer positioned to access a [−150, 200] meV energy window around the elastic line. All measurements utilized a high-heat-load diamond (1110) monochromator and an additional Si(844) upstream monochromator was used for the high-resolution measurements in order to obtain an overall energy transfer resolution of 20 meV. To minimize Thomson scattering the $2\theta$ angle was fixed at $90°$. This led to a small variation of the $L$ values between 5.7 and 6 r.l.u. An iris analyzer mask was used to achieve a spectrometer momentum resolution of $\pm 0.048\,Å^{-1}$. RIXS data was collected using a photon-counting CdTe 2D detector tilted along the energy dispersion direction to provide an effective pixel size of $25\,\mu m \times 55\,\mu m$. Data was converted from pixels to energy by applying a calibration factor of $epp = 4.5$ meV/pixel. The uncertainty in determining $E = 0$ meV using incoherent scattering from an elastic line standard corresponds to $\pm 1$ pixel. Each spectra shown are the average of 2–3 scans.

### Exact diagonalization calculations

Crystal field excitations in $H_3LiIr_2O_6$ and $\alpha$-$Li_2IrO_3$ were modeled by considering a Hamiltonian including spin-orbit coupling ($\lambda = 540$ meV) in the large cubic crystal field limit $H = H_U + H_{CF} + H_t$ including on-site Coulomb interactions ($H_U$), trigonal distortions $H_{CF}$ and nearest-neighbors hopping $H_t$. $H_t$ follows the form described in ref. [57] and ref. [55]. The leading hopping integrals are $t_\parallel$, that parameterizes hopping between parallel orbitals and $t_O$ that encodes hopping paths mediated by O $2p$ orbitals. These two hopping integrals were sampled from a normal distribution with standard deviations $\sigma_{t_0} = 50$ meV, $\sigma_{t_\parallel} = 10$ meV. For the magnitude of the trigonal fields, $\delta$, the width of the normal distribution was allowed to vary between $H_3LiIr_2O_6, \sigma_\delta = 20$ meV, and $\alpha$-$Li_2IrO_3, \sigma_\delta = 5$ meV[55]. Our calculations explore the phase diagram given by the mean values ($\delta, t_O, t_\parallel$) to find the best agreement with the data. For $H_3LiIr_2O_6$ we found $\delta = -47$ meV, $t_O = 440$ meV, and $t_\parallel = -50$ meV. For $\alpha$-$Li_2IrO_3$ this was $\delta = -50$ meV, $t_O = 400$ meV, and $t_\parallel = -30$ meV (see Supplementary Information).

### O-K edge XAS

Low-temperature O-K edge XAS measurements were performed at the beamline 8 of the ALS in both partial fluorescence yield and total electron yield. We fit the data to a combination of two Lorentzians to account for the $\alpha$ peak, two Gaussians for the $\beta$ peak and an arctan step.

## Data availability

The RIXS data generated in this study have been deposited in the Zenodo database under accession code 8161522[60].

## Code availability

The code used in this study is available from the authors upon reasonable request.

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

## Acknowledgements

We thank J. Knolle and P. A. Lee for useful discussions. Work performed at Brown University by A.d.l.T., B.Z., and K.P. was supported by the U.S. Department of Energy, Office of Science, Office of Basic Energy Sciences, under Award Number DE-SC0021265. Work carried out at Boston College by F.B. and F.T. was supported by the National Science Foundation under award number DMR-2203512. This research used resources of the Advanced Photon Source, a U.S. Department of Energy (DOE) Office of

Science User Facility operated for the DOE Office of Science by Argonne National Laboratory under Contract No. DE-AC02-06CH11357. This research used resources of the Advanced Light Source, which is a DOE Office of Science User Facility under contract no. DE-AC02-05CH11231.

## Author contributions

A.d.l.T. and B.Z. performed the X-ray spectroscopy measurements with support from M.H.U., J.K., G.F., G.-H.L., W.Y., and D.H. on samples synthesized by F.B. and F.T. A.d.l.T. and K.W.P. analyzed and interpreted the data. A.d.l.T. and K.W.P. wrote the paper with input from all authors.

## Competing interests

The authors declare no competing interests.
