## [Peer Review File · Nature Communications]

Reviewers' Comments:

Reviewer #1:

Remarks to the Author:

The authors present an interesting study on the magnetic state of H3LiIr2O6 using RIXS. The results showed a broad bandwidth and momentum-independent continuum of magnetic excitations with the center around 25 meV at low temperatures, and the authors suggested that H3LiIr2O6 is a disordered topological spin liquid in close proximity to bond-disordered versions of the Kitaev quantum spin liquid. The experimental data appears to be solid, and the underlying physics is intriguing.

However, I have some concerns regarding the manuscript that need to be addressed before I can recommend it for publication in Nature Communications. Firstly, the authors assume that RIXS directly probes the dynamical spin-spin correlations of the single spin-flip process. While the authors compared the experimental results with theoretical predictions for the spin-spin correlation function, no justification was provided as to why this assumption holds. Furthermore, the manuscript mentions a spectral weight transfer between multiple modes, suggesting that multiple spin-flip processes may be involved in RIXS measurements. Disentangling the RIXS cross-section into different spin-flip channels is challenging, and if RIXS measures the two-spin-flip process, as in Raman scattering, the momentum-independent magnetic continuum in H3LiIr2O6 would not be surprising. Therefore, the authors should provide a justification for their assumption.

Secondly, the authors only present the momentum-dependent dispersion along the high symmetry direction at 10 K. It is unclear how the results differ at high temperatures, e.g., $T=300$ K. While a-Li2IrO3 and H3LiIr2O6 may have significantly different magnetic states at low temperatures, they should have similar magnetic responses at room temperature. Therefore, the authors should provide results at high temperatures to support their conclusion that H3LiIr2O6 is not a disorder limited correlated remnant of a-Li2IrO3.

Once these issues have been addressed, I would be happy to re-evaluate the manuscript for publication in Nature Communications.

Reviewer #2:

Remarks to the Author:

This manuscript reports on resonant inelastic x-ray spectroscopy (RIXS) on the layered iridate compound H3LiIr2O6 belonging to a prominent class of honeycomb iridates being possible candidates for a Kitaev-type quantum spin-liquid (KQSL) ground state. The strong frustration, which is active in the title compound can be characterized by the absence of long-range magnetic order down to the lowest temperatures despite a relatively high Curie-Weiss temperature of almost 100 K. However, the system is prone to disorder effects, mainly due to a high density of stacking faults of the 2d honeycomb planes, but does not undergo static spin-glass freezing. In addition, zero-point motion of hydrogens linking the honeycomb layers also may play a role in suppressing long-range magnetic order. Hence it seems important to disentangle disorder effects from the expected Kitaev physics. This is attempted in the present manuscript.

Critically one could state that even in the cleanest samples of proximate KQSL, in recent literature there exists a never-ending controversy about possible hallmarks of Kitaev physics. Certainly, strong disorder adds a further important variable, which can hardly be controlled and quantified. Both, Kitaev interactions and disorder tend to suppress long-range magnetic order and it seems hard to disentangle these effects. However, there exist a number of theoretical models trying to combine disorder with Kitaev exchange. Despite the complexity of the material, I understand that it seems important to finally identify a Kitaev spin liquid with fractionalized excitations and to prove the importance of disorder. In this respect the present manuscript is a step into the right direction. However, I see several weak points and unresolved questions which have to be solved before publication.

The main observation of these experiments is an almost momentum independent broad continuum centered around 25 meV with a width of approximately 40 meV. The momentum independence speaks against conventional magnetic fluctuations close to some type of magnetic long-range

order. It seems to be dictated by strong disorder (with no static spin-glass freezing) and possibly from Kitaev-type interactions. However, I do not see clear fingerprints of Kitaev physics. This could be probably hidden in the temperature dependence. But the temperature dependence is not well represented. Fig. 2d and 2e show spectra at 10 and 300 K, with little changes (slight stiffening and broadening on increasing temperature). Assuming that Kitaev physics is absent at high temperatures (well above the Curie-Weiss temperature) the room-temperature spectra probably represent the purely paramagnetic response. With decreasing temperature, fractionalized excitations should come into play, and this should be hidden in the continuum response. Hence, I think that the temperature dependence should be documented and discussed in much more detail. It also would be interesting to see the evolution of high energy tail (Fig. 1b) with increasing temperatures. In addition, a closer comparison with the results of the continuum observed in Raman experiments (Ref. 21) is definitively needed.

The continuum excitation allows a rough estimate of the Kitaev exchange, and this value should be compared in more detail with estimates from other experiments and with those from theory.

Minor comments:

In explaining experimental details, the authors state that high-resolution RIXS experiments offer an alternative to neutron scattering techniques. However, the present experiments obviously were performed with state-of-the art resolution.

The authors should provide a statement, if phonon contributions can be completely neglected. Fig. 1d) and e): The authors should provide an explanation for the strongly temperature dependent elastic line.

Summary: Please include references to the statement that bond-disorder KSQL models are the closest in describing the present experimental observation.

Reviewer #3:

None

Reviewer #1 (Remarks to the Author):

The authors present an interesting study on the magnetic state of H3LiIr2O6 using RIXS. The results showed a broad bandwidth and momentum-independent continuum of magnetic excitations with the center around 25 meV at low temperatures, and the authors suggested that H3LiIr2O6 is a disordered topological spin liquid in close proximity to bond-disordered versions of the Kitaev quantum spin liquid. The experimental data appears to be solid, and the underlying physics is intriguing.

=====
We thank the reviewer for carefully reading our manuscript and for recognizing the fascinating nature of the low-energy magnetic excitations of H3LiIr2O6.
=====

However, I have some concerns regarding the manuscript that need to be addressed before I can recommend it for publication in Nature Communications. Firstly, the authors assume that RIXS directly probes the dynamical spin-spin correlations of the single spin-flip process. While the authors compared the experimental results with theoretical predictions for the spin-spin correlation function, no justification was provided as to why this assumption holds.

=====
The reviewer is correct to point out that a direct quantitative mapping of the RIXS cross-section to the single spin flip contribution of the dynamical structure factor $S(Q,\omega)$ remains a subject of active research. However, for the specific case of octahedrally coordinated Iridium based 5d5 Mott insulators, the dominant contribution of the single spin-flip process to the RIXS cross-section has now been well established, both theoretically [Phys. Rev. B 96, 085108 (2017)], and experimentally [Physical Review X 10, 021034 (2020), Nature Physics 11, 462–466 (2015) Physical Review Research 2, 043094 (2020)]. In particular, the single spin-flip process has been demonstrated to dominate the RIXS response of alpha-Li2IrO3 and Na2IrO3 [Physical Review X 10, 021034 (2020), Nature Physics 11, 462–466 (2015) Physical Review Research 2, 043094 (2020)]. While higher-order processes are also present in the RIXS cross-sections for all of these compounds and in H3LiIr2O6, the extensive experimental and theoretical results provide a litmus test to the one-to-one correspondence of RIXS intensity and $S(Q,\omega)$. We have clarified the relationship between $S(Q,\omega)$ and RIXS in the main text.
=====

Furthermore, the manuscript mentions a spectral weight transfer between multiple modes, suggesting that multiple spin-flip processes may be involved in RIXS measurements.

We thank the referee for pointing out a possible point of confusion. We have performed a more detailed temperature dependence of the low energy excitations in H3LiIr2O6, shown here and in Figure 2 of the main text. The center position of the DHO follows the trend of the magnetic susceptibility showing a softening as temperature is reduced near the temperature onset of magnetic correlation [PHYSICAL REVIEW B 92, 115122

(2015), PHYSICAL REVIEW B 93, 174425 (2016)] (~200K), as signified by the hump in the susceptibility. Given our energy resolution, we cannot distinguish if the observed temperature dependence corresponds to the transfer of spectral weight between low and high energy modes below and above CW or to a mode softening that follows the temperature trend of the magnetic susceptibility. [Ref 19 of the main text, Nature 554, 341–345 (2018)]. We have changed the language in the main text and added a more detailed discussion of the temperature dependence.

Disentangling the RIXS cross-section into different spin-flip channels is challenging, and if RIXS measures the two-spin-flip process, as in Raman scattering, the momentum-independent magnetic continuum in H3LiIr2O6 would not be surprising. Therefore, the authors should provide a justification for their assumption.

As we discussed above, we agree with the referee that disentangling the different contributions to the RIXS cross-section is challenging and beyond the scope of this paper. However, the dominant contribution of $S(Q,\omega)$ to the direct RIXS cross-section for octahedrally coordinated Ir⁴⁺ compounds has by now been well established. In ordered Honeycomb Kitaev magnets, the low energy excitation energy and intensity, as measured by RIXS, agree with the single spin flip cross-section as predicted by linear spin wave calculations [Phys. Rev. B 96, 085108 (2017), Physical Review X 10, 021034 (2020), Nature Physics 11, 462–466 (2015) Physical Review Research 2, 043094 (2020)]. Since the RIXS process will be exactly the same in H3LiIr2O6, our measurements must also predominantly reflect $S(Q,\omega)$ in our title compound. Moreover, as theoretically discussed in Phys. Rev. Lett. 117, 127203 (2016) the non-spin conserving channels (NSC) map into $S(Q,\omega)$ as measured by INS and dominate the cross-section, while the spin conserving (SC) would be more closely related to what Raman measures have a much smaller contribution. Without polarization analyzing optics, the RIXS intensity is dominated by NSC and can be related to $S(Q,\omega)$.

Higher-order processes (multimagnons) are probably present, but their cross-section is expected to be weaker. Additionally, the intensity variation of the multimagnons signal will reflect a joint density of states of the underlying magnons and exhibit some characteristic momentum dependence [see Phys. Rev. Lett. 117, 127203 (2016)]. This disagrees with our observation of a continuum of magnetic excitations at low temperature of a momentum-independent intensity, characteristic of a lack of translational symmetries.

The role of disorder in the Kitaev-Heisenberg Hamiltonian, in general, and the origin of the low energy excitations in H3LiIr2O6, in particular, remain a matter of debate [See references 31-36 and 48 of the main text]. Unveiling the energy and momentum dependence, or lack of, the collective excitations in H3LiIr2O6 is fundamental to understanding the interplay of exchange parameters and disorder, a problem of fundamental importance to understanding quantum materials. In this context, our observation of a momentum-independent continuum is interesting and not trivial.

We also note a recently published Phys. Rev. Research 5, 023009 (2023), [48] discusses how the inclusion of disorder in the Kitaev model in proximity to the ferromagnetic Kitaev QSL leads to drastic variations of $S(Q,\omega)$ upon slight variations of the magnetic exchanges, supporting our findings. Our results provide a clear microscopic observable for theorists tackling this important challenge. They are consistent with recent literature that points toward a disordered state in close proximity to the KQSL.

=====

Secondly, the authors only present the momentum-dependent dispersion along the high symmetry direction at 10 K. It is unclear how the results differ at high temperatures, e.g., $T=300$ K. While α -Li₂IrO₃ and H₃Li₂Ir₂O₆ may have significantly different magnetic states at low temperatures, they should have similar magnetic responses at room temperature. Therefore, the authors should provide results at high temperatures to support their conclusion that H₃Li₂Ir₂O₆ is not a disorder limited correlated remnant of α -Li₂IrO₃.

=====

We thank the reviewers for their suggestions for more detailed temperature dependence. As a testament to the importance of our results, we obtained a limited amount of additional beam time before the Advanced Photon Source extended shutdown to perform additional temperature-dependence measurements. These are now shown in Fig. 2 of the revised manuscript. At room temperature, we observe overdamped excitations in a broad continuum that does not change at any of the three high symmetry points we measured (Fig 2c). As the referee suggests, our RIXS data is consistent with previous room-temperature RIXS measurements in Na₂IrO₃ and α -Li₂IrO₃ [Phys Rev Lett 2, 043094 (2020)] as expected for the paramagnetic states of all three compounds. The detailed temperature evolution of the magnetic excitations we now present demonstrates that the momentum-independent continuum we observe at low temperatures is distinct from the room-temperature paramagnetic excitations.

=====

Once these issues have been addressed, I would be happy to re-evaluate the manuscript for publication in Nature Communications.

=====

We hope the reviewer finds our reply satisfactory and that they appreciate our significant additional efforts in collecting new data and majorly revising our manuscript. We hope they can recommend our manuscript for publication in Nat. Comms.

=====

Reviewer #2 (Remarks to the Author):

This manuscript reports on resonant inelastic x-ray spectroscopy (RIXS) on the layered iridate compound $\text{H}_3\text{LiIr}_2\text{O}_6$ belonging to a prominent class of honeycomb iridates being possible candidates for a Kitaev-type quantum spin-liquid (KQSL) ground state. The strong frustration, which is active in the title compound can be characterized by the absence of long-range magnetic order down to the lowest temperatures despite a relatively high Curie-Weiss temperature of almost 100 K. However, the system is prone to disorder effects, mainly due to a high density of stacking faults of the 2d honeycomb planes, but does not undergo static spin-glass freezing. In addition, zero-point motion of hydrogens linking the honeycomb layers also may play a role in suppressing long-range magnetic order. Hence it seems important to disentangle disorder effects from the expected Kitaev physics. This is attempted in the present manuscript.

Critically one could state that even in the cleanest samples of proximate KQSL, in recent literature there exists a never-ending controversy about possible hallmarks of Kitaev physics. Certainly, strong disorder adds a further important variable, which can hardly be controlled and quantified. Both, Kitaev interactions and disorder tend to suppress long-range magnetic order and it seems hard to disentangle these effects. However, there exist a number of theoretical models trying to combine disorder with Kitaev exchange. Despite the complexity of the material, I understand that it seems important to finally identify a Kitaev spin liquid with fractionalized excitations and to prove the importance of disorder. In this respect the present manuscript is a step into the right direction.

=====
We thank the reviewer for their in-depth summary of the field, the critical reading of our paper, and for understanding the importance of our work. RIXS is the only technique able to resolve the low-energy excitations of a fascinating compound, $\text{H}_3\text{LiIr}_2\text{O}_6$, and provide a new understanding of the interplay of disorder and Kitaev interactions.
=====

However, I see several weak points and unresolved questions which have to be solved before publication.

The main observation of these experiments is an almost momentum independent broad continuum centered around 25 meV with a width of approximately 40 meV. The momentum independence speaks against conventional magnetic fluctuations close to some type of magnetic long-range order. It seems to be dictated by strong disorder (with no static spin-glass freezing) and possibly from Kitaev-type interactions. However, I do not see clear fingerprints of Kitaev physics. This could be probably hidden in the temperature dependence. But the temperature dependence is not well represented. Fig. 2d and 2e show spectra at 10 and 300 K, with little changes (slight stiffening and broadening on increasing temperature). Assuming that Kitaev physics is absent at high temperatures (well above the Curie-Weiss temperature) the room-temperature spectra probably represent the purely paramagnetic response. With decreasing temperature, fractionalized excitations should come into play, and this should be

hidden in the continuum response. Hence, I think that the temperature dependence should be documented and discussed in much more detail. It also would be interesting to see the evolution of high energy tail (Fig. 1b) with increasing temperatures.

=====

We thank the referee for this comment. Following their suggestion, we returned to Sector 27 of the APS to complete a more detailed temperature dependence (Right and Fig. 2 of the revised manuscript). Figure 2a shows the RIXS intensity at a Gamma point ($L = 5.7$) as a function of temperature. The solid line is a fit to the data. The shaded color-coded area encodes the damped harmonic oscillator (DHO). As we observed previously, the circular markers in Figure 2b show that the DHO center hardens with increasing temperature above T_{CW} . The center position of the DHO follows the trend of the magnetic susceptibility showing an initial

hardening near the temperature onset of magnetic correlation [PHYSICAL REVIEW B 92, 115122 (2015), PHYSICAL REVIEW B 93, 174425 (2016)] ($\sim 200K$), as signified by the hump in the susceptibility. Near room temperature three times above CW, the DHO is centered at 35 meV as shown in our initial measurements.

=====

In addition, a closer comparison with the results of the continuum observed in Raman experiments (Ref. 21) is definitively needed.

=====

Ref 21 shows Raman intensity at various temperatures in $H3LiIr2O6$. They observe a continuum of magnetic intensity that peaks at around 40 meV. Assuming that the magnetic Raman response is associated with creating two pairs of fractionalized excitations, this will correspond to a $K \sim 26$ meV, in excellent agreement with the extracted value from our RIXS data. The temperature dependence of the integrated Raman susceptibility decreases as the temperature is lowered. This opposes $\alpha\text{-Li2IrO3}$ and $\alpha\text{-RuCl3}$, in which the same quantity decreases with temperature. This was interpreted as resulting from a dominant two-spin flip process in $H3LiIr2O6$, which directly probes any existent fractionalized excitation in this compound.

We plotted in Figure 2b (square markets) the extracted amplitude of the DHO from a fit to the data, normalized to the $T = 10$ K value. We observe an increase in the amplitude with a lowering of the temperature. Note that the RIXS cross-section is dominated by the single spin flip processes in $S(Q,w)$, while Raman is sensitive to two-spin-flip processes [Phys. Rev. Lett. 117, 127203 (2016)]. A direct comparison between the RIXS and Raman crosssection is complex, demands polarization analysis for the RIXS measurements, and out of the scope of this work, the observed temperature dependence resembles that observed by Raman in other

Kitaev honeycomb candidates [Phys. Rev. B 95, 174429 (2017); Phys. Rev. B 101, 174436 (2020)] providing strong evidence for a low-temperature correlated state in $\text{H}_3\text{LiIr}_2\text{O}_6$ distinct from a paramagnet at room temperature.

The continuum excitation allows a rough estimate of the Kitaev exchange, and this value should be compared in more detail with estimates from other experiments and with those from theory.

We thank the referee for this suggestion. The main text now states: "While a random distribution of exchange values due to disorder might hinder the exact determination of K , the extracted value from our analysis $K \sim 25$ meV, is comparable to that of $\alpha\text{-Li}_2\text{IrO}_3$ and Na_2IrO_3 [13–16, 50], to the value extracted from the maximum of the continuum intensity in Raman scattering measurements [21], and to that from first principles calculations [26, 27]"

Minor comments:

In explaining experimental details, the authors state that high-resolution RIXS experiments offer an alternative to neutron scattering techniques. However, the present experiments obviously were performed with state-of-the-art resolution.

We thank the referee for this suggestion and noting that our measurements represent the state-of-the-art. We have clarified the main text to say: "Although resonant inelastic X-ray scattering (RIXS) remains limited by state-of-the-art energy resolution comparable to the theoretically predicted magnitude of K , high-resolution RIXS experiments at the Ir L₃ edge have emerged as an alternative technique to access $S(\mathbf{q}, \omega)$ in 5d Mott insulators [13, 14, 16, 45]"

The authors should provide a statement, if phonon contributions can be completely neglected.

We have added a statement to the main text " This signal resonates at the Ir L₃ edge, confirming the direct RIXS nature of these excitations and further ruling out a phonon origin [16] (see Supplementary information)", as well as a discussion in section S1.

Fig. 1d) and e): The authors should provide an explanation for the strongly temperature dependent elastic line.

We thank the referee for pointing this out. Our first initial temperature dependence was at two slightly different Q positions near the $[0,0,6]$ structural Bragg peak. The T dependence reported now was taken under more controlled experimental conditions, with the sample being carefully realigned at each temperature step. Not obvious temperature dependence is now seen at the elastic line.

Summary: Please include references to the statement that bond-disorder KSQL models are the closest in describing the present experimental observation.

=====
References have been added to the concluding paragraph.
=====

Reviewers' Comments:

Reviewer #1:

Remarks to the Author:

After thoroughly reviewing the previous reports and the responses provided by the authors, I still have concerns regarding the momentum dependence of the resonant inelastic X-ray scattering (RIXS) spectra in H3LiIr2O6 at room temperature. In Figure 2c of the manuscript, the authors present RIXS spectra at Gamma, K, and M points. They argue that there is no discernible momentum dependence in the intensities, which they claim is consistent with previous high-temperature RIXS measurements in Na2IrO3 and $\alpha\text{-Li2IrO3}$. However, I find this claim questionable, as it appears to contradict the findings reported in Ref. 16.

Ref. 16 presents results in Figure 9, which demonstrate that both $\alpha\text{-Li2IrO3}$ and Na2IrO3 exhibit momentum-dependent intensities, indicating the presence of nearest-neighbor spin-spin correlations—a key property of the Kitaev model. The authors' findings for H3LiIr2O6 seem inconsistent with these observations, raising doubts about the presence of nearest-neighbor spin-spin correlations in H3LiIr2O6 and its connection to the physics of the Kitaev model. For those disordered Kitaev models, although the bonds are disordered, the local spin-spin correlations should remain at least at room temperature.

Moreover, the momentum independence observed in the RIXS spectra of H3LiIr2O6 at room temperature appears to be at odds with the main results claimed in the manuscript. The manuscript highlights this momentum-independent feature as a key aspect, suggesting its relevance to the Kitaev physics. However, this feature is already present at room temperature and does not seem to be directly related to the Kitaev quantum-spin-liquid physics which only plays important roles at low temperatures. Additionally, the behavior of the DHO mode is mentioned in the report, indicating that it only displays softening below 105 K without significant differences from the paramagnetic state. These raise concerns about the validity of the manuscript's main selling point.

Considering the aforementioned concerns, I am unable to recommend the publication of this manuscript in Nature Communications until the authors adequately address these issues. The presence of momentum-independent RIXS spectra in H3LiIr2O6 at room temperature challenges the main claims made in the manuscript and raises doubts about the connection between the observed magnetic excitations and the physics of the Kitaev model.

Reviewer #2:

Remarks to the Author:

This is the revised version of a manuscript reporting on resonant inelastic x-ray spectroscopy on the layered iridate compound H3LiIr2O6 belonging to the prominent class of honeycomb iridates being possible candidates for a Kitaev-type quantum spin-liquid. The authors significantly revised the manuscript answering almost all questions and remarks of the reviewers in detail. I think that this revised version can be published in its present form in Nature Communications.

Reviewer #1 (Remarks to the Author):

After thoroughly reviewing the previous reports and the responses provided by the authors, I still have concerns regarding the momentum dependence of the resonant inelastic X-ray scattering (RIXS) spectra in $\text{H}_3\text{LiIr}_2\text{O}_6$ at room temperature. In Figure 2c of the manuscript, the authors present RIXS spectra at Γ , K, and M points. They argue that there is no discernible momentum dependence in the intensities, which they claim is consistent with previous high-temperature RIXS measurements in Na_2IrO_3 and $\alpha\text{-Li}_2\text{IrO}_3$. However, I find this claim questionable, as it appears to contradict the findings reported in Ref. 16.

Ref. 16 presents results in Figure 9, which demonstrate that both $\alpha\text{-Li}_2\text{IrO}_3$ and Na_2IrO_3 exhibit momentum-dependent intensities, indicating the presence of nearest-neighbor spin-spin correlations—a key property of the Kitaev model. The authors' findings for $\text{H}_3\text{LiIr}_2\text{O}_6$ seem inconsistent with these observations, raising doubts about the presence of nearest-neighbor spin-spin correlations in $\text{H}_3\text{LiIr}_2\text{O}_6$ and its connection to the physics of the Kitaev model. For those disordered Kitaev models, although the bonds are disordered, the local spin-spin correlations should remain at least at room temperature.

We thank the reviewer for their comment, but we respectfully disagree that our data contradicts that of Figure 9 in Ref 16. In their important and informative work, Revelli et al. show sinusoidal modulations in the RIXS intensity that varies across multiple Brillouin Zones (BZ). Such slow variation over a broad reciprocal space range is due to nearest neighbor limited spin-spin correlations. It is important to remark that this intensity modulation depends on the $\cos^2(q \cdot r/2)$ with “r” a vector parameterizing the Ir-Ir bonds [equations (1),(2), and (3) in Ref. 16]. As such, this intensity variation is only apparent for measurements spanning many Brillouin zones and vanishes in the limit of $q \cdot r \rightarrow 0$. This is shown for $\alpha\text{-Li}_2\text{IrO}_3$ in panels c and d of Ref. 16, Fig. 9, reproduced here for clarity [Fig 1 b and c]. The observed intensity modulation periodicity is $6 \cdot (2\pi/a)$ along the Γ -M direction and $\sim 8.4(2\pi/a)$ along Γ -K. In Figure 2c of our manuscript (reproduced here in Fig 1 a), we show momentum dependence within a single Brillouin Zone ($G = [0,0]; H=[0.5,0.5], K=[2/3,1/3]$), which corresponds to a very small range of the q space explored in Ref. 16 as highlighted by the red vertical bar in panels Fig 1 (c) and (d).

Although we observe a $\sim 10\%$ modulation, this is within our error bars which prevent us from discussing the $\cos^2(q.r/2)$ dependence in more detail.

We agree with the referee that measuring the RIXS intensity over multiple BZ at room temperature would provide another comparison point to the phenomenology of Kitaev magnets $\alpha\text{-Li}_2\text{IrO}_3$ and Na_2IrO_3 , in which Kitaev interactions are known to be present but not dominant at low temperatures. However, this was infeasible given the limited amount of additional beamtime granted to us to perform temperature-dependent measurements before the APS shutdown. Our conclusions are fully supported by the existing data, and the observed intra-BZ intensity homogeneity is consistent with Ref 16.

We have modified the text to clarify that our momentum dependence is within a BZ in Figure 2c and to situate the momentum dependence of Figure 3 in the context of Ref 16.

=====

Moreover, the momentum independence observed in the RIXS spectra of $\text{H}_3\text{LiIr}_2\text{O}_6$ at room temperature appears to be at odds with the main results claimed in the manuscript. The manuscript highlights this momentum-independent feature as a key aspect, suggesting its relevance to the Kitaev physics. However, this feature is already present at room temperature and does not seem to be directly related to the Kitaev quantum-spin-liquid physics which only plays important roles at low temperatures.

=====

The referee brings up an important and challenging question of how one can distinguish between the continuum from a paramagnet and that of a fractionalized state in the presence of disorder. This is indeed a key question in the field, and our manuscript empirically addresses it. $\text{H}_3\text{LiIr}_2\text{O}_6$ currently stands as the only candidate Kitaev magnet absent any frozen moment or glassy physics, making it an important model material to study the interplay of disorder and dominant nearest-neighbor Kitaev-like interactions.

As the referee correctly points out, the observation of momentum-independent excitations does not signify dominant Kitaev interactions but is a consequence of the combination of the intrinsic disorder, present at all temperatures, and Kitaev interactions in $\text{H}_3\text{LiIr}_2\text{O}_6$. As an example, we append in Figure 2 a calculation for $S(Q)$ for the Kitaev Heisenberg model without (top row) and with bond disorder Kitaev mode (bottom row) adapted from Ref 52 [Phys. Rev. Research 5, 023009 (2023)]. The figure shows how $S(Q)$ changes as a function of the relative value of K (Kitaev Exchange) and J (Heisenberg interactions) as the ground state evolves from a ferromagnetically ordered state to a zig-zag phase via the KQSL state. In the Kitaev phase, $S(Q)$ shows intra-BZ intensity modulations that reflect the sign of the Kitaev interaction. However, including random bond disorder leads to modifications of the momentum dependence of the energy-integrated intensity. This is shown in the bottom row where disorder redistributes the spectral weight resulting in a more homogenous $S(Q)$ across BZ. We remark that the disorder in the model is just an approximation to the complex intrinsic disorder in $\text{H}_3\text{LiIr}_2\text{O}_6$.

Nevertheless, our data are most consistent with the disordered Kitaev limit. Thus, we conclude that the magnetic state in H3LiIr2O6 results from dominant Kitaev interaction and the intrinsic disorder present at all temperatures.

We thank the referee for allowing us to clarify our message. We acknowledge that the discussion of the origin of the continuum and the momentum of independence could be improved. We have modified the main text accordingly, as well as the bold paragraph, which now states: “ *In the low-temperature correlated state, we uncover a broad bandwidth of magnetic excitations. The center energy and high-energy tail of the continuum are consistent with expectations for dominant ferromagnetic Kitaev interactions between dynamically fluctuating spins. However, the absence of a momentum dependence to these excitations indicates a disorder-induced broken translational invariance*”.

Additionally, the behavior of the DHO mode is mentioned in the report, indicating that it only displays softening below 105 K without significant differences from the paramagnetic state.

Once more, the referee highlights a main challenge in the search for quantum spin liquid phases: how to distinguish between thermally fluctuating spins in a paramagnetic state and correlated dynamically fluctuating spins. Given that broad excitation in energy and momentum characterize the excitation spectrum in both states, a detailed temperature dependence across Curie-Weiss (TCW) is needed. We thank the referees again for their suggestions during the initial round of reviews that led us to collect the data presented in Figure 2b (black markers),

reproduced here for clarity. The observation of a softening of the magnetic excitation continuum in H3LiIr2O6 across the key energy scale for the onset of magnetic correlations (T_{cw}) undoubtedly signifies that the correlated magnetic state in H3LiIr2O6 is distinct from the paramagnetic state.

=====

These raise concerns about the validity of the manuscript's main selling point.

Considering the aforementioned concerns, I am unable to recommend the publication of this manuscript in Nature Communications until the authors adequately address these issues. The presence of momentum-independent RIXS spectra in H3LiIr2O6 at room temperature challenges the main claims made in the manuscript and raises doubts about the connection between the observed magnetic excitations and the physics of the Kitaev model.

=====

We thank the referee again for reading our manuscript in detail and pointing out possible points of confusion. We urge the referee to consider our reply, our revised manuscript, and the opinion of Reviewer #2. We respectfully request them to change their position towards the publication of our manuscript in Nature Communications. Our data reveal the existence of magnetic excitations at high energies, centered at 25 meV and extending up to $E \approx 170$ meV. These two characteristic energy scales are suggestive of calculations of $S(q, \omega)$ for pure, extended, and bond-disorder gapless KQL. Additionally, the center energy is consistent with the energy scale extracted in Raman measurements for H3LiIr2O6 and the Kitaev exchange of α -Li2IrO3 and Na2IrO3. Our work demonstrates the presence of dominant Kitaev interactions on a disordered background and paves the way to understanding the $T = 0$ nature of H3LiIr2O6. As this material currently stands as the only candidate Kitaev compound not exhibiting any spin freezing or static moments as $T \rightarrow 0$, it is a highly important and topical model material. By revealing the full excitation spectrum of this state, our work informs future theoretical work on disorder models for quantum spin liquids. It demonstrates the complex interplay of Kitaev physics and disorder.

Reviewer #2 (Remarks to the Author):

This is the revised version of a manuscript reporting on resonant inelastic x-ray spectroscopy on the layered iridate compound $\text{H}_3\text{LiIr}_2\text{O}_6$ belonging to the prominent class of honeycomb iridates being possible candidates for a Kitaev-type quantum spin-liquid. The authors significantly revised the manuscript answering almost all questions and remarks of the reviewers in detail. I think that this revised version can be published in its present form in Nature Communications.

=====

We thank the referee for their careful consideration of their work and for recommending it for publication in Nature Communications.

Reviewers' Comments:

Reviewer #1:

Remarks to the Author:

The authors thoroughly addressed all the issues and revised the manuscript accordingly. Based on its current form, I recommend the publication of this manuscript in Nature Communications.

Reviewer #1 (Remarks to the Author):

The authors thoroughly addressed all the issues and revised the manuscript accordingly. Based on its current form, I recommend the publication of this manuscript in Nature Communications.

We thank the Reviewer for their careful read of our manuscript and for recommending it for publication in Nature Communications.